# Scrubber Filter in the Phosphate Fertilizer Factory Reduces Fluorine Emission and Accumulation in Corn

**Gleidson Junior Silva and Risely Ferraz-Almeida ***

Luiz de Queiroz College of Agriculture, University of São Paulo, Piracicaba 13418-900, São Paulo, Brazil
* Correspondence: rizely@gmail.com

**Abstract:** Fluorine (F) produced from the fertilizer factory occurs in the process of phosphate fertilizer production, using sulfur and phosphate rocks as raw materials. Technologies to control atmospheric pollution with F should be adopted to reduce the impact on agricultural production. This study has the hypothesis that the emission of F, derived from the chimneys of fertilizer factories, is influencing the quality of corn (*Zea mays* L.) and increasing the F levels in the soil and plants. The objective of the study was to monitor the contents of F in corn leaves and soil in properties located close to the fertilizer production industry (between 1.5 and 2.0 km) before and after the installation of scrubber filters in the chimneys of the factory. A field study was carried out during the 2020/2021 harvest to evaluate the contents of F in corn plants and soil. Results showed that the scrubber filter installation represented a F reduction average of 92% in leaves comparing the average before the scrubber filter installation. Corn showed symptoms of F toxicity, such as leaf chlorosis, caused by the disintegration of chloroplasts, inhibition of photosynthesis, and others. In addition, there was a reduction of 40% (from the first to the second collecting) and 75% (from the first to the third collecting) in the levels of F in the soil after the scrubber filter installation. Based on the results, we conclude that the implementation of a scrubber filter is an optimal alternative to reduce F levels in corn leaves and the soil in properties located close to a fertilizer factory.

**Keywords:** phosphate; *Zea mays* L.; phytotoxicity; pollution; atmosphere





## 1. Introduction

Fluorine (F) is an element with the highest reactivity among all non-metallics, reacting with numerous organic and inorganic substances and forming stable compounds with aluminum. F is commonly found in the environment with a F ion and is estimated at 0.077% of the total content in the Earth's crust [1,2]. The sources of F contamination can be both natural and/or anthropogenic [3,4]. Natural sources are mainly the activities of volcanoes and their emanations, rocks, and minerals (such as apatite, cryolite, fluorite, and topaz), which lead to F accumulations in soil, oceans, lakes, rivers, and other forms of natural water [5,6]. Endemic fluorosis has the highest incidence in China, India, and various African regions [7]. According to the World Health Organization [8], most of the contaminated waters present a F content above 1.5 mg dm$^{-3}$.

In Brazil, the main anthropogenic sources of F are industrial activities that release F into the atmosphere, which are followed by the production of phosphate fertilizers [9]. The production of fertilizers comes from burning phosphate rocks (such as apatite and fluorite) with the release of gaseous F in chimneys into the atmosphere through mists [10]. In the process, phosphate rocks are treated with sulfuric acid for acidulation to make phosphorus more available in fertilizer [11,12]. The fluoride is released into the atmosphere in possible failures in the gas-washing process in the plant or due to other possible failures that may occur during the process. Intermediate products (such as nitric acid and phosphoric acid) and basic fertilizers (such as diammonium phosphate, monoammonium phosphate, triple superphosphate, and simple superphosphate) are obtained and subsequently transformed

into final fertilizers [13]. The F impact on the environment has been noticed relatively recently and is related directly to the industrial activity, production of fertilizers, and aluminum smelters with the F emission into the atmosphere [14].

When in contact with plants or soils, the F is relatively immobile in soil and can cause severe effects on plant development. Under natural conditions, the soil F levels are lower than 1 mg L$^{-1}$. However, in regions with anthropogenic releasing, the F levels can reach 10 mg L$^{-1}$. Wenzel and Blum [2], evaluating contaminated soils (luvisol and regosol) with F released in metallurgical regions, concluded that the solubility of total F was influenced by soil conditions (pH, organic matter, and contents of aluminum). The Brazilian soils are characterized by low contents of organic matter and nutrients (calcium, magnesium, phosphorus, and others) and low pH due to natural weathering [15,16]. Mirlean et al. [17] showed that in a region of Rio Grande, located in the estuarine area of Lagoa dos Patos (Rio Grande do Sul; Brazil), the presence of F in the surface layer of the soil was associated with the propagation of air contamination emitted by factories of fertilizers.

In plants, F has not been considered an essential element with a concentration generally lower than 10 μg F/g dry weight in most species [18]. F is considered one of the most toxic elements for plants with a possible yield reduction of up to 50% for highly sensitive plants (i.e., wheat; *Triticum* L.) [19,20]. The symptoms of F phytotoxicity are the wrinkling and necrotic regions of the leaves with a reduction in absorbing solar radiation [21]. F gas can penetrate the leaf tissue of plants, most often through the stomata or through the cuticle and lenticels of branches. It is noteworthy that a part of this F is mobile and soluble; therefore, repeated rains lead to a decrease in the concentration of this pollutant [22].

Fluorine is also known to inhibit the activity of antioxidant enzymatic systems (such as superoxide) interfering with cell signaling. Symptoms of fluoride toxicity in plants also include a reduction in growth and development, abscission of leaves, flowers, and fruits, and reduction in seed production [23]. In seed germination and seedling growth, F is known to prevent phosphorylation of the phytin compound in tissues through inhibition of the phytase enzyme, promoting a reduction in the growth rate of seedling roots during germination [24,25].

In the field, some factors may influence the amount of fluorine deposited in plants due to external agents and their emissions, such as proximity to the source and wind speed and direction [22]. It is known that some plants perform better than others on F tolerance; specifically, corn (*Zea mays* L.) is a plant that does not have a tolerance for F and when recommended for silage can influence animal nutrition due to the high amount of F in plant tissue. In Brazil, corn is used in feeding ruminants, poultry, and pigs, directly influencing human food and promoting food and nutritional security [26,27]. Therefore, studies are needed to understand the impact of F levels in corn leaves that are used for animal feed.

Currently, several atmospheric pollution control technologies are being studied, such as the scrubber filter that is installed to collect all the dust that comes out of industrial processes. This system is considered simple to operate and from time to time receives compressed air jets and counterflow to clean these filters.

This study has the hypothesis that the emission of F, derived from the chimneys of fertilizer factories, is influencing the quality of corn and increasing the F levels in the soil and plants. The objective of the study was to monitor the contents of F in corn leaves and soil in properties located close to the fertilizer production industry before and after the installation of scrubber filters in the chimneys of the factory.

## 2. Materials and Methods

### 2.1. Region Characterization

The study was carried out in the Córrego do Sal region, Araxá, Minas Gerais, Brazil (9°35′36″ South; 46°56′27″ West), during one harvest in 2020/2021 (November to March). The Araxá region is located in the Triângulo Mineiro Region, 368 km from Belo Horizonte, the state capital (Minas Gerais), as shown in Figure 1. The region presents an average annual temperature between 18 and 22.7 °C and rainfall indices of around 1525 mm.

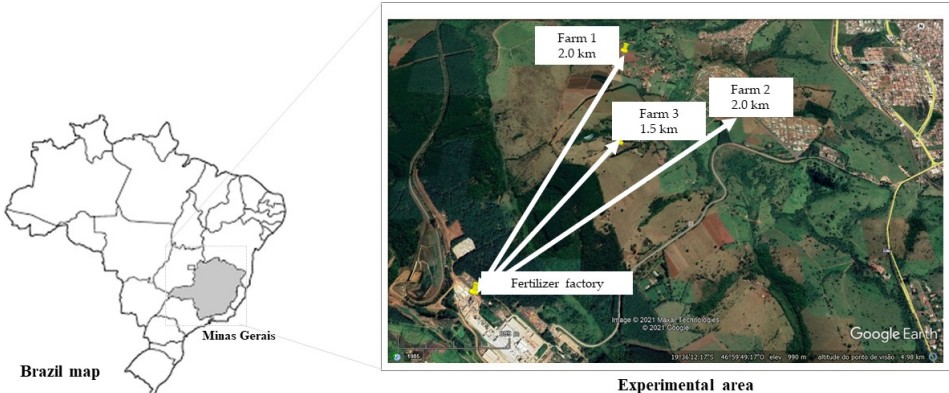

**Figure 1.** Study area in the Córrego do Sal region, Araxá, Minas Gerais, Brazil, during the 2020/2021 harvest. Distances between the fertilizer factory and Farm 1 (2.0 km), Farm 2 (2.0 km), and Farm 3 (1.5 km).

The region of Araxá has great food production with an emphasis on potato (*Solanum tuberosum*), coffee (*Coffea* sp.), soybean (*Glycine max*), and corn. In addition, the region of Araxá stands out for the exploitation of rocks and minerals, such as phosphate, which is used in the production of fertilizers. In addition, there is the exploitation of niobium, a rare and abundant mineral, used in metallic alloys.

The study was developed in a region close to a phosphate fertilizer factory monitoring the impact of scrubber filter installation in local agriculture. Scrubber filters are atmospheric emissions treatment systems developed to carry out the retention and neutralization of gaseous chemical components. They use liquids to wash, cool, or react with polluting substances contained in process gases to be released into the atmosphere, recycled, or in fuel gases used in other parts of the process (UTBR®). The scrubber filters present a cylindric format, stainless steel, 304-vertical model, and support high pressures (Figure 2).

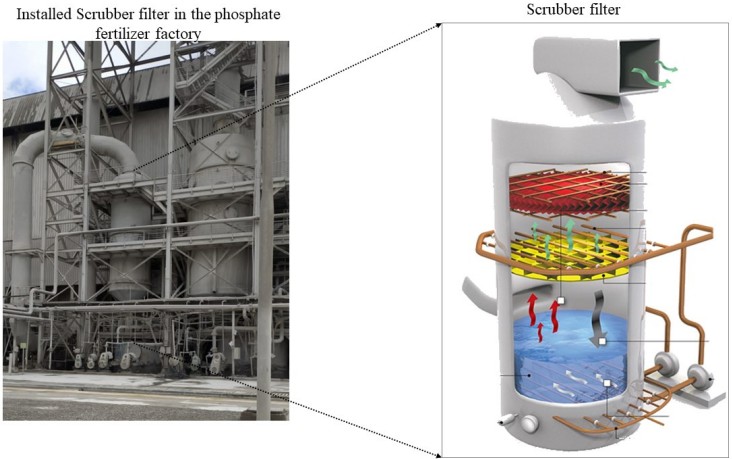

**Figure 2.** Scrubber filter in the phosphate fertilizer factory located in the Córrego do Sal region, Araxá, Minas Gerais, Brazil.

## 2.2. Sample Collecting

Data collection was carried out in three rural properties, located in the region of Araxá, with distances of 2.0 (Farm 1), 2.0 (Farm 2), and 1.5 km (Farm 3) from the phosphate fertilizer factory (Figure 1). These rural properties are classified as small, with about 10 hectares, in which the farms produce their silage (from corn and pasture) to feed the cattle on their properties. Each property has about 15 to 30 half-blood cattle ¾ and 7/8, which spend most of the year in free pasture areas, and in the dry season depending on the silage produced on the property.

In each farm, leaves were collected (opposite and below the ear) at the height of the ear insertion at the end of each plant cycle. Collections were manually performed with a total of 30 samples for the area. The first collection was carried out 30–50 days after planting, the second was carried out 60–70 days after planting, and the last collection was carried out 80–90 days after planting. The corn harvest was performed 100–110 days after planting. The second and third collecting were performed after the installation of the scrubber filter in the fertilizer factory.

The leaf samples were sent to the IAC (Agronomic Institute of Campinas) to monitor the contents of F in the leaf, according to the methodology described by the IAC [28]. During the study, there was no application of pesticides or foliar fertilizers in the area. Symptoms of fluoride toxicity in plants were monitored in the first collection, which was carried out 30–50 days after planting. The data represent the result of one crop cultivation during the 2020/2021 (November to March) harvest.

Soil samples were also collected on rural properties to monitor the F contents in the laboratory, according to the NBR ISO/IEC 17025–CRL 0354 [8]. In each area, 20 soil samples were collected with a zigzag pattern with a total of 5 points for each farm which was considered as replications. There were no additional soil and leaf samples due to the clear reduction after the scrubber filter installation during one harvest in 2020/2021 (November to March). Points close to termites, anthills, houses, roads, corrals, animal manure, fertilizer deposits, limestone, or soil stains were avoided.

### 2.3. Data Analysis

The descriptive statistics, assumptions of normality (Shapiro–Wilk test), and homogeneity of variance (Bartlett test) were tested using a $p$ of 0.05. When present, the outliers were identified and excluded by the Grubbs test.

Soil and leaf samples in each property were treated as replicates for data analysis. The averages of the F in the soil and leaves before and after the implementation of the scrubber filter were compared according to Student's $t$-test ($t$-test; $p < 0.05$).

Variables were correlated by the Pearson correlation test using a probability of 0.05. A relation between F in leaves and soil (*RLS*) was calculated using Equation (1), indicating that there was an accumulation of F in the soil for each unit of F in leaves

$$RLS = \frac{FS}{FL} \tag{1}$$

where *RLS* is the relation between F in leaves and soil; *FS*: F in soil; *FL*: F in leaves. Statistical analysis was performed in R (version 4.0.0; R Foundation for Statistical Computing), and the results were graphed in SigmaPlot (version 11.0; SYSTAT Software, Inc.).

### 3. Results

#### 3.1. F in Leaves and Soil

Before the scrubber filter installation, the content of F in leaves of corn was 124.0 mg kg$^{-1}$, with a respective reduction after the scrubber filter installation and averages of 5.9 and 4.6 mg kg$^{-1}$ at 60–70 and 80–90 days after the planting (general average of 5.0 mg kg$^{-1}$), respectively. The scrubber filter installation represented a F reduction average of 92% in leaves comparing the average before and after (Figure 3).

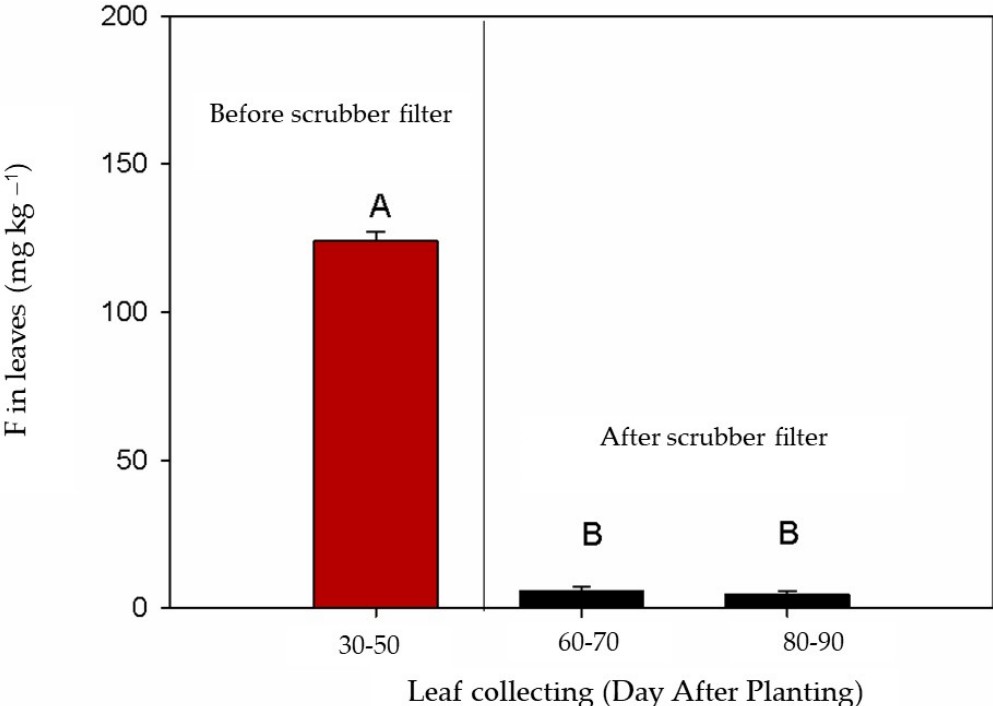

**Figure 3.** Fluorine (F) in leaves of corn in rural properties located close to a fertilizer factory in the Araxá region, Minas Gerais, Brazil. Leaves were collected 30–50 days after planting (before the scrubber filter) and 60–70 and 80–90 days after planting (after the scrubber filter). Averages were compared according to Student's *t*-test (*t*-test; $p < 0.05$), and significant results were represented by different uppercase letters (A and B) added in the bars. The study was developed during one harvest in 2020/2021 (November to March).

Before the scrubber filter installation, there were symptoms of fluoride toxicity in plants with chlorosis, necrosis, and abscission of leaves. Plants presented a height average of 1.83 cm with 60% of plants with symptoms of fluoride toxicity (Figure 4).

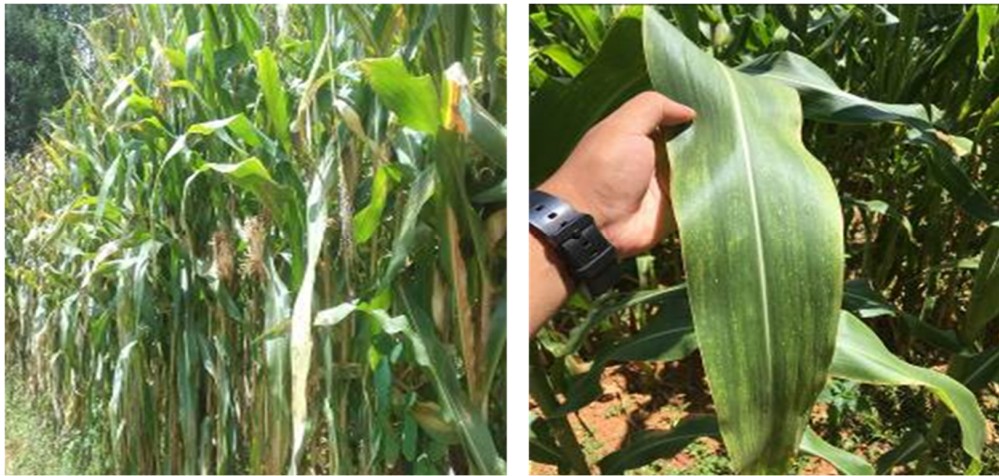

**Figure 4.** Symptoms of fluoride toxicity in plants of corn in rural properties located close to a fertilizer factory in the Araxá region, Minas Gerais, Brazil. Plants were monitored in the first collection (30–50 days after planting). The study was developed during one harvest in 2020/2021 (November to March).

In soil, the F content was 211.9 mg kg$^{-1}$ after 30–50 days of planting. After the scrubber filter installation, there was a reduction in the contents of F in soil with a respective average of 130.0 and 54.3 mg kg$^{-1}$ 60–70 and 80–90 days after the planting (Figure 5). The scrubber filter installation represented a reduction of 40% (from the first to the second collecting) and 75% (from the first to the third collecting) of F in soil (Figure 5).

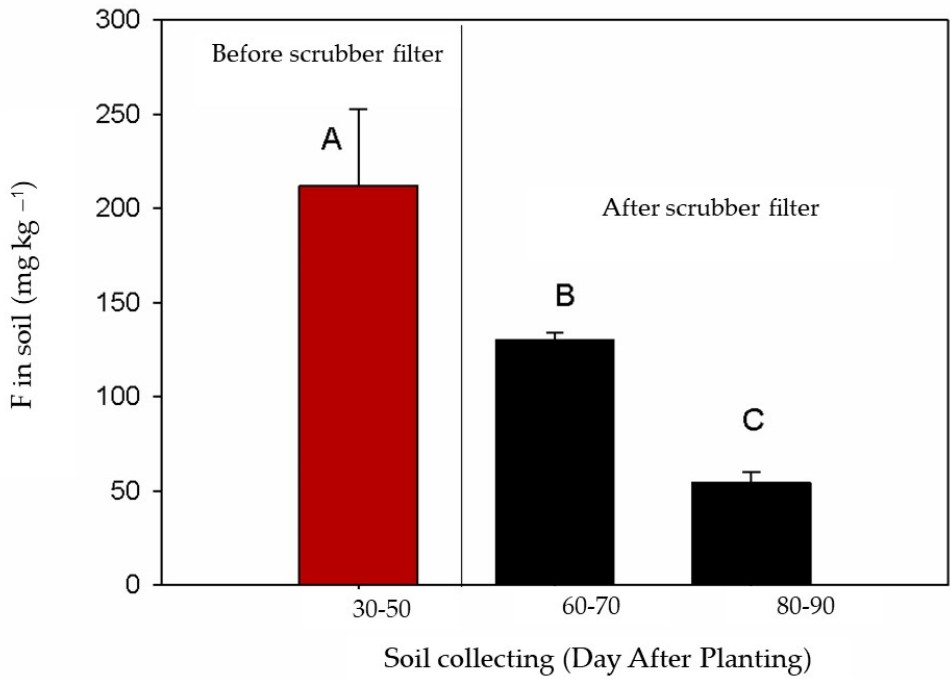

**Figure 5.** Fluorine (F) in soil cultivated with corn in rural properties located close to a fertilizer factory in the Araxá region, Minas Gerais, Brazil. The soil was collected 30–50 days after planting (before the scrubber filter) and 60–70 and 80–90 days after planting (after the scrubber filter). Averages were compared according to Student's *t*-test (*t*-test; $p < 0.05$), and significant results were represented by different uppercase letters (A, B, and C) added in the bars. The study was developed during one harvest in 2020/2021 (November to March).

*3.2. Corn Yield*

Corn yield ranged from 2.6 to 5.7 t ha$^{-1}$ in the farms with higher yield: Farms 1 and 2. Farm 3 presented the lowest corn yield, representing a reduction of 52% compared with the average of Farms 1 and 2 (5.3 t ha$^{-1}$), as shown in Table 1. Farm 3 presented a lower distance from the fertilizer factory with a total of 2.0 km.

**Table 1.** Corn yield (t ha$^{-1}$) in rural properties (Farm 1, 2, and 3) located close to a fertilizer factory in the Araxá region, Minas Gerais, Brazil.

| Farms | Corn Yield |
| --- | --- |
| | t ha$^{-1}$ |
| Farm 1 | 5.7 ± 3.2 A |
| Farm 2 | 5.0 ± 0.1 A |
| Farm 3 | 2.6 ± 0.5 B |

Averages were compared according to Student's *t*-test (*t*-test; $p < 0.05$), and significant results were represented by different uppercase letters (A and B) added in the column.

Distances were measured between the fertilizer factory and Farm 1 (2.0 km), Farm 2 (2.0 km), and Farm 3 (1.5 km). Averages were compared according to Student's *t*-test (*t*-test; $p < 0.05$), and significant results were represented by different uppercase letters (A and B) added with the average of each farm. The study was developed during one harvest in 2020/2021 (November to March).

Before the scrubber filter, the *RLS* was higher than 0.5 with a reduction between 0.03 and 0.06 with the installation of the scrubber filter. This result indicates that for each unit of F in the soil, there was a higher accumulation of F in leaves before the installation of the scrubber filter (Figure 6).

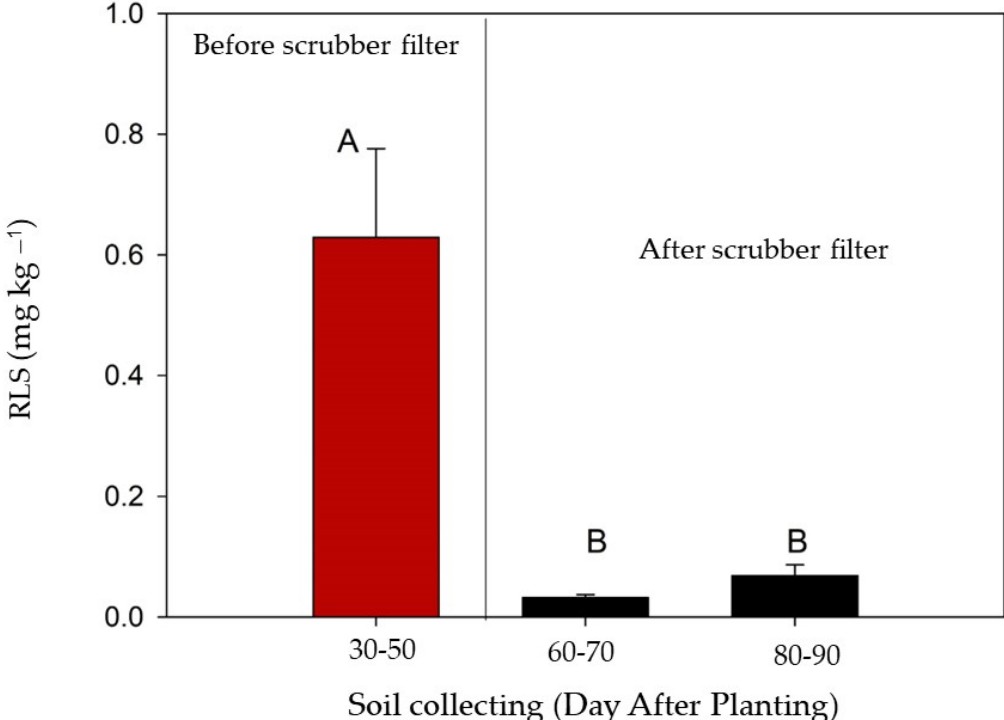

**Figure 6.** Relation between F in leaves and soil (*RLS*) in plants of corn in rural properties located close to a fertilizer factory in the Araxá region, Minas Gerais, Brazil. Plants were monitored in the first collection (30–50 days after planting). Averages were compared according to Student's *t*-test (*t*-test; $p < 0.05$), and significant results were represented by different uppercase letters (A and B) added in the bars. The study was developed during one harvest in 2020/2021 (November to March).

There was a significant correlation between F in soil and leaves with an r of 0.74 ($p < 0.05$), indicating that the increase in F in soil increased the F in leaves. However, there was no significant correlation between corn yield with F in soil (r: −0.14; $p > 0.05$) and leaves (r: 0.10; $p > 0.05$).

## 4. Discussion

After the installation of the scrubber filter, there were reductions in F contents in corn leaves and soil. These reductions are important and demonstrate that the addition of a scrubber filter is an optimal alternative to reduce the F emission from the fertilizer industry. The impact of F derived from industry on soil, water, and plants has been presented by other studies [29,30], with a negative effect on crop yield and located populations [31]. Rizzu et al. [3] demonstrated that F accumulation in food can cause nutritional surveys among the populations, in which children and teenagers are the categories considered at major risk.

Leaf F contents higher than 100 mg kg$^{-1}$ are considered toxic for the development of plants during the vegetative phase with a direct influence on the absorption of nutrients from the soil [29,30]. The interpretation of the F toxicity in plants varies according to species, cultivar, and the stage of development of the plants. Generally, the F levels in plant materials from areas not exposed to polluting sources are below 10 mg kg$^{-1}$ of dry matter, which are values close to those found in leaves after scrubber filter implantation [31].

Corn is classified as a plant sensitive to F, like other plants (gladiolus, colonião grass, eucalyptus, grape, guava), while plants with intermediate tolerance are mango, wheat, eucalyptus grandis, citrus, and soybean. Gladioli are considered a plant highly sensitive to F and can become necrotic with 20 mg kg$^{-1}$ of F (20 mg F/g of mass drought), while cotton plants can be seen to be diseased with more than 4000 mg kg$^{-1}$ of F [32]. In China, studies have explained the F uptake and accumulation in tea plant (*Camellia sinensis* (L.) O. Kuntze), which is well known as one F hyper-accumulator [5].

In our study, the plants presented symptoms of fluoride toxicity with chlorosis, necrosis, and abscission of leaves before the installation of a scrubber filter with an average of 124.0 mg kg$^{-1}$ of F in leaves. According to Brewer et al. [32], F values ranging from 29 to 48 mg kg$^{-1}$ are enough to reduce corn production in field conditions. Plant F accumulation causes visible injuries with biochemical changes impacting the enzyme activities (such as ATP synthase, ribulose biphosphate carboxylase oxygenase), sucrose synthase, and the content of chlorophyll. Chloroplasts and mitochondria are the main sites of F accumulation [33]. Plants tend to accumulate F mainly in the root system and be dose-dependent with some exceptions [3]. Fortes [34] also noticed symptoms of chlorosis interveinal in the leaves of corn with an intensity of symptoms in the filling stage of grains in an area of production located close to a ceramic industry, in São Paulo, Brazil. Pelc et al. [35] showed a decrease in germination, inhibition of root growth, and inhibition of catalase activity in embryos and roots of wheat cultivars with the increase in fluoride concentration on plastic Petri dishes with a NaF solution.

The visible injuries in leaves occur because the fluorine gas penetrates the plants through the stomata and accumulation in areas close to the point of entry, where the greatest damage occurs [36,37]. Mackowiak et al. [38] in a hydroponic experiment showed a high F level in rice (3217 mg F− kg$^{-1}$ DM) demonstrating a hyper-accumulation in the shoot (395 mg F− kg$^{-1}$ DM). Sugarcane also presented with a F hyper-accumulation with values of 521 and 120 mg F− kg$^{-1}$ DM (sugar cane) in roots and leaves, respectively [39]. In corn, the F is accumulated mainly in roots and leaves with the greatest value in roots (542 mg F− kg$^{-1}$ DM) [40,41].

In soil, F contents showed a greater amount before the implementation of improvements in the fertilizer factory's production system, and it also was reduced as seen in the reductions in the amount of F in the leaves. In the soil, according to the parameters of the IAC [28], there are no studies for parameters of the critical level of F in the soil. In soil, the F can move due to the high mobility of atmospheric fluorine compounds that are highly soluble (i.e., fluoride) and easily reach the water table [31]. Mirlean et al. [17] noticed that in regions poor in organic matter and clay minerals (dystrophic sandy soils), there is a higher concentration of fluoride in groundwater, which is characterized by the level of atmospheric contamination. Wenzel and Blum [2] showed that the F solubility was influenced by soil pH, where the higher F solubility was noticed in higher soil acidity. Brazilian soils are commonly acidic due to natural weathering with high organic matter oxidation and phosphorus immobilization [42–44]. Therefore, the risk of contamination to the food chain and the water table is low in low-acid soils, but it increases in strongly acidic conditions as well as in alkaline conditions (cultivated soil).

The soil F contents were observed in farms that are close to the factory (between 1.5 and 2.0 km from the farms and fertilizer factory), which explained the high F level in plants and soil. Sokolova et al. [30] demonstrated contaminated soils 0.5 km from an aluminum industry in Russia with a direct effect on productivity and F uptake of crops. Meanwhile, Mirlean et al. [17] noticed the increase in soil F at a distance of 2.5 km from the factories. Assumpção et al. [45] also demonstrated that corn plant exposure to atmospheric F with a distance between 1 and 2 km from the ceramic industry reduced plant growth and grain production. In our study, F contents in soil and plants influenced the corn yield mainly in Farm 3, which is closest to the factory (2.6 t ha$^{-1}$). There was a significant correlation between F in soil and leaves; however, there was no significant influence of F in soil and leaves with corn yield. These results indicated that the increase in F in soil increased the F

in leaves, but it did not influence the yield. In Farm 3, the corn yield was considered 40% lower when compared to the Brazilian annual productivity, which presented an average of 4 t ha$^{-1}$ [27–47]. Probably, the high F mobility in soil influenced the soil collecting and the absence of correlation in our study.

Generally, in the area close to the factory, there is financial compensation due to anomalies that may cause intoxication in the environment. In this condition, the farmers are rewarded for their proximity to the fertilizer plant. The use of a scrubber filter based on results demonstrated to be an optimal alternative to reduce the F impact from fertilizer factories. It is an important outcome because Brazil is an important corn producer with around 60% of the Brazilian agricultural production located in Cerrado (where the study was developed) [48–50].

## 5. Conclusions

The installation of a scrubber filter in the phosphate fertilizer factory reduces fluorine impact in corn areas with lower F contents in soil (a decrease of 75% compared to the average before and after the scrubber filter) and leaves of corn (a decrease of 92% comparing the average before and after the scrubber filter). Before the scrubber filter installation, there were symptoms of fluoride toxicity in plants with chlorosis, necrosis, and abscission of leaves. There were no additional soil and leaf samples due to the clear reduction in F after the scrubber filter installation during one harvest in 2020/2021. However, periodic monitoring should be performed by the industry to monitor the continuous efficiency of the scrubber filter. Our results conclude that the use of a scrubber filter can be an optimal alternative to reduce the F impact from fertilizer factories in local agriculture areas. It is an important outcome because Brazil is an important corn producer with around 60% of the Brazilian agricultural production located in Cerrado.

**Author Contributions:** Conceptualization, G.J.S. and R.F.-A.; data curation, G.J.S. All authors have read and agreed to the published version of the manuscript.

**Funding:** This research received no external funding.

**Institutional Review Board Statement:** Not applicable.

**Informed Consent Statement:** Not applicable.

**Data Availability Statement:** Not applicable.

**Acknowledgments:** Thanks to the University of São Paulo, Luiz de Queiroz College of Agriculture (USP/ESALQ), the Coordenação de Aperfeiçoamento de Pessoal de Nível Superior (CAPES; grant number 88882.317567/2019-01), the Agronomic Institute of Campinas (IAC), and the fertilizer Factory in Araxá, Minas Gerais.

**Conflicts of Interest:** The authors declare no conflict of interest.

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
