# Peer review of "Scrubber Filter in the Phosphate Fertilizer Factory Reduces Fluorine Emission and Accumulation in Corn"

_2813-4168, doi:10.3390/air1010005_

Round 1

Reviewer 1 Report

Scrubber filter in the phosphate fertilizer factory reduces fluorine emission and accumulation in corn - REVIEW

The study presents the effects of the use of scrubbers in a fertilizer factory to reduce the atmospheric pollution due to fluorine emissions from the factory. These effects were quantified by the amounts and effects of Fluorine on corn plants and the soil close to the factory, before and after the implementation of the scrubbers. The studies indicated a reduction in the quantity of Flouring in the plants, and soil.

This is an interesting study. The authors should, please, address the issues highlighted below that could enhance their manuscript.

ABSTRACT

Line 15

“…located close to the fertilizer production industry…”

How close to the factory?

Lines 20 to 21

The scrubber filter installation represented a reduction average of 92% comparing the average before and after the scrubber filter installation.

What was this 92% reduction for? The level of Fluorine in the soil, or on the corn leaves, or the emissions from the factory chimneys?

Lines 23 to 25

However, after the maintenance and installation of equipment, there was a reduction in the levels of F in the plants. This reduction of F was also verified in the soil after the implantation of the scrubber filter.

Please quantify the reduction levels mentioned here.

-        - - -

The authors should give a brief review of recent similar studies that have looked into the reduction of Fluorine or other emissions from a fertilizer factory and their effects on soil and crop contamination. This is lacking in this study.

MATERIALS AND METHODS

Line 94

“…during 2019/20 and 2020/21 harvests.”

What months or specific periods are these harvests?

Line 102

In Figure 1, please add a scale to the map to help the reader to understand the distances between the farms and the Fertilizer factory.

RESULTS

Line 198

“there was symptoms of fluoride toxicity in”

This should change to “there were symptoms of fluoride toxicity in”

Line 240

“soil there was an higher accumulation of in leaves before the installation of scrubber filter”

This sentence does not make sense, please re-write it.

Lines 304 to 305

“Generally, in area close to Factory there is a financial compensation due to anomalies that may cause intoxication in the environmental.”

What is “financial compensation”? Are farmers rewarded for their proximity to the fertilizer plant?

-        - - -

It is not clear if the results [Figures 3, 5, 6] are averages from both the 2019/20 and 2020/21 seasons, and how each season differed from each other.

The distances of each of the farms from the fertilizer factory and their potential effect on the results should be discussed.

How do the results compare to previous recent studies that looked at this subject?

Author Response

REVISOR #1

REVISOR #1: Scrubber filter in the phosphate fertilizer factory reduces fluorine emission and accumulation in corn – REVIEW. The study presents the effects of the use of scrubbers in a fertilizer factory to reduce atmospheric pollution due to fluorine emissions from the factory. These effects were quantified by the amounts and effects of Fluorine on corn plants and the soil close to the factory, before and after the implementation of the scrubbers. The studies indicated a reduction in the quantity of Flouring in the plants, and soil. This is an interesting study. The authors should, please, address the issues highlighted below that could enhance their manuscript.

AUTHORS: Dear Reviewer, thank you for the suggestions. We followed all comments and added them in our manuscript. All editions are described here and added in the manuscript in green.

REVISOR #1: ABSTRACT: Line 15 “…located close to the fertilizer production industry…” How close to the factory? Lines 20 to 21; The scrubber filter installation represented a reduction average of 92% comparing the average before and after the scrubber filter installation. What was this 92% reduction for? The level of Fluorine in the soil, on the corn leaves, or the emissions from the factory chimneys? Lines 23 to 25. However, after the maintenance and installation of equipment, there was a reduction in the levels of F in the plants. This reduction of F was also verified in the soil after the implantation of the scrubber filter. Please quantify the reduction levels mentioned here.

AUTHORS: all suggestions were added in the abstract to give a clear idea in the manuscript.

REVISOR #1: The authors should give a brief review of recent similar studies that have looked into the reduction of Fluorine or other emissions from a fertilizer factory and their effects on soil and crop contamination. This is lacking in this study.

AUTHORS: The introduction was added based on your comments and other reviewers. Thank you for the recommendations.

REVISOR #1: MATERIALS AND METHODS Line 94, “…during 2019/20 and 2020/21 harvests.” What months or specific periods are these harvests? Line 102. In Figure 1, please add a scale to the map to help the reader to understand the distances between the farms and the Fertilizer factory.

AUTHORS: We added the information of the months and the distance between the farms and the factor.

REVISOR #1: RESULTS. Line 198: “there were symptoms of fluoride toxicity in”; This should change to “there were symptoms of fluoride toxicity in”; Line 240 “soil there was a higher accumulation of in leaves before the installation of scrubber filter” This sentence does not make sense, please re-write it. Lines 304 to 305 “Generally, in the area close to Factory there is a financial compensation due to anomalies that may cause intoxication in the environmental.” What is “financial compensation”? Are farmers rewarded for their proximity to the fertilizer plant?

AUTHORS: We added the editions in the manuscript and explained the financial compensation in the text.

REVISOR #1: It is not clear if the results [Figures 3, 5, 6] are averages from both the 2019/20 and 2020/21 seasons, and how each season differed from the other.

AUTHORS: we added that information in the Material and Methods. We are presenting crop cultivation.

REVISOR #1: The distances of each of the farms from the fertilizer factory and their potential effect on the results should be discussed. How do the results compare to previous recent studies that looked at this subject?

AUTHORS: The distances of each of the farms were added in the text and Figure 1. We also explained this information in the Discussion.

Reviewer 2 Report

I have read the manuscript "Scrubber filter in the phosphate fertilizer factory reduces fluorine emission and accumulation in corn". The study monitored the accumulation of fluorine in corn leaves and soil in properties close to the fertilizer production industry and the efficiency of scrubber filters in phosphate fertilizer production.

Although a certain amount of applied basic research has been carried out in this paper, it can still not meet the requirements of publishing in Air journal. The main reasons are as follows:

1. The paper topic lacks innovation and needs to condense key scientific or technical issues.

2. In the introduction, there is too little content.

3. In the result section, there is a lack of corresponding figures and tables as support.

4. The amount of original data in the paper is too small, and the content needs greatly expanded.

5. This paper has many grammatical errors and obvious low-level mistakes. It is recommended to check carefully, or you can ask a professional organization to revise.

6. The content of the main part of the whole big thesis is too thin.

7. It is recommended to continue to do in-depth topics.

Author Response

REVISOR #2

REVISOR #2: I have read the manuscript "Scrubber filter in the phosphate fertilizer factory reduces fluorine emission and accumulation in corn". The study monitored the accumulation of fluorine in corn leaves and soil in properties close to the fertilizer production industry and the efficiency of scrubber filters in phosphate fertilizer production. Although a certain amount of applied basic research has been carried out in this paper, it can still not meet the requirements of publishing in an Air journal. The main reasons are as follows: 1. The paper topic lacks innovation and needs to condense key scientific or technical issues.2. In the introduction, there is too little content. 3. In the result section, there is a lack of corresponding figures and tables as support. 4. The amount of original data in the paper is too small, and the content needs greatly expanded. 5. This paper has many grammatical errors and obvious low-level mistakes. It is recommended to check carefully, or you can ask a professional organization to revise. 6. The content of the main part of the whole big thesis is too thin. 7. It is recommended to continue to do in-depth topics.

AUTHORS: Dear Reviewer, thank you for the suggestions. We followed all comments and added them in our manuscript. All editions are described here and added in the manuscript in green.

Reviewer 3 Report

Please see the attached PDF file.

Author Response

REVISOR #3

REVISOR #3: Review of “Scrubber filter in the phosphate fertilizer factory reduces fluorine emission and accumulation in corn” by Gleidson Junior Silva et al. Major comment: This manuscript conducted a field study to monitor the accumulation of Fluorine in corn leaves and soil in farms located close to the fertilizer production industry to investigate the effect of scrubber filter on reducing the levels of Fluorine in both plants and soil. It was found the implementation of scrubber filter promoted the reduction of Fluorine evidently. This study indicates the application of filtration system is necessary to reduce the adverse effect of air pollutant on the environment. The minor questions below need to be addressed before the manuscript can be published.

AUTHORS: Dear Reviewer, thank you for the suggestions. We followed all comments and added them in our manuscript. All editions are described here and added in the manuscript in green.

REVISOR #3: Minor questions: 1. What do the “A”, “B”, “C” mean in Figure 3, 5 and 6? Is the data in these figures based on one farm or the average of three farms? Is there any difference between the data for different farms?

AUTHORS: Perfect observation, we added a text with information about the letters.

REVISOR #3: What is the effect of distance between the farm and the industry region on the concentration of Fluorine in the leaves and soil? What is the effect of this distance on the performance of scrubber filter in reducing the concentration of Fluorine?

AUTHORS: The distances of each of the farms were added in the text and Figure 1. We also explained this information in the Discussion.

REVISOR #3: 2. More parameters about the scrubber filter are suggested to be added to the manuscript (i.e., volume flow rate, cross section area)

AUTHORS: More information about the scrubber filter was added in the Material and Methods.

Round 2

Reviewer 1 Report

THE MANUSCRIPT HAS BEEN IMPROVED.

PLEASE CONSIDER THE COMMENTS BELOW.

Scrubber filter in the phosphate fertilizer factory reduces fluorine emission and accumulation in corn – REVIEW 2

ABSTRACT

Lines 23 to 25

However, after the maintenance and installation of equipment, there was a reduction in the levels of F in the plants. This reduction of F was also verified in the soil after the implantation of the scrubber filter.

Please quantify the reduction levels mentioned here.

MATERIALS AND METHODS

Line 110

In Figure 1, “factor” should be “factory.”

RESULTS

It is not clear if the results [Figures 3, 5, 6] are averages from both the 2019/20 and 2020/21 seasons, and how each season differed from each other.

Author Response

REVIEW 2 (Editions in green in the manscript)

REVIEW 2: ABSTRACT Lines 23 to 25: However, after the maintenance and installation of equipment, there was a reduction in the levels of F in the plants. This reduction of F was also verified in the soil after the implantation of the scrubber filter. Please quantify the reduction levels mentioned here.

AUTHORS: The information of reductions was added. Thank you for all your comments.

REVIEW 2: MATERIALS AND METHODS: Line 110 In Figure 1, “factor” should be “factory.”

AUTHORS: It was made, thanks.

REVIEW 2: RESULTS:  It is not clear if the results [Figures 3, 5, 6] are averages from both the 2019/20 and 2020/21 seasons, and how each season differed from the others.

AUTHORS: The study was developed during one harvest in 2020/21 (November to March) from 2020 to 2021.  We added that information in the Figures to elucidate this condition.